# Association of Polymorphisms in the Long Non-Coding RNA *HOTAIR* with Recurrent Pregnancy Loss in a Korean Population

**DOI:** 10.3390/genes13112138

**Published:** 2022-11-17

**Authors:** Hyeon Woo Park, Young Ran Kim, Jeong Yong Lee, Eun Ju Ko, Min Jung Kwon, Ji Hyang Kim, Nam Keun Kim

**Affiliations:** 1Department of Biomedical Science, College of Life Science, CHA University, Seongnam 13488, Republic of Korea; 2CHA Bundang Medical Center, Department of Obstetrics and Gynecology, School of Medicine, CHA University, Seongnam 13496, Republic of Korea

**Keywords:** recurrent miscarriage, recurrent spontaneous abortion, long non-coding RNA, single nucleotide polymorphism, *HOTAIR*

## Abstract

Recurrent pregnancy loss (RPL) affects 1% to 5% of women, with devastating effects on both reproductive health and psychological well-being. Homeobox (*HOX*) transcript antisense RNA (*HOTAIR*) is a long non-coding RNA (lncRNA) produced by *HOXC*; it plays a major role in invasion and development of ovarian and other cancers. The aim of the present study was to analyze effects of *HOTAIR* polymorphisms (rs4759314 A>G, rs920778 T>C, rs1899663 G>T, and rs7958904 G>C) on RPL in Korean women. A total of 403 women with RPL and 383 healthy women were selected for this study. Genotyping analysis was performed with the polymerase chain reaction, restriction fragment length polymorphism, and the TaqMan genotyping assay. Clinical characteristics were compared using Student’s unpaired *t*-test and the chi-square test for categorical variables. Logistic regression was performed to evaluate associations between single nucleotide polymorphisms and RPL incidence. In all assays, *p* < 0.05 was considered significant. *HOTAIR* polymorphisms rs4759314A>G and rs920778T>C were highly associated with increased risk of RPL, specifically the haplotypes rs4759314A>G/rs1899663G>T (G-T) and rs4759314A>G/rs920778 T>C (G-C). These associations were maintained in haplotypes that contained three polymorphisms (rs4759314 A>G, rs920778 T>C, and rs1899663 G>T) A-C-G, G-T-G, and G-T-T, further indicating that the *HOTAIR* rs4759314 and rs920778 polymorphisms play significant roles in idiopathic RPL in Korean women.

## 1. Introduction

Recurrent pregnancy loss (RPL), also referred to as recurrent miscarriage or recurrent spontaneous abortion, is defined as two or more consecutive miscarriages that occur before 20 weeks of gestation [1,2,3]. RPL is a significant clinical problem that affects up to 5% of reproductively active women and can cause emotional and psychological distress [4,5].

A total of 10–12% of pregnant women experience pregnancy loss within 8–12 weeks of gestation, and the incidence of RPL is 1% of all pregnancies [6,7]. RPL is a polygenic disease; fetal chromosomal abnormalities, maternal factors, thrombophilia, endocrine disorder, immune dysfunction, smoking, stress, and genetic variation can be causes of RPL [3,8].

The homeobox C (*HOXC*) gene is a member of the *HOX* superfamily, which plays an essential role during embryonic development. Long non-coding RNAs (lncRNAs), which are non-coding RNA transcripts > 200 nucleotides in length, are associated with epigenetic abnormalities in different cellular events [9,10]. *HOX* transcript antisense RNA (*HOTAIR*) is an lncRNA product of *HOXC* that has been implicated in invasion and progression of various cancers [11,12], with high levels of *HOTAIR* reported in papillary thyroid carcinoma and epithelial ovarian cancer [13].

*HOTAIR* is located in *HOXC* cluster genes, which are located in chromosome 12 and regulate *HOXD* genes [14]. Recently, *HOTAIR* was shown to play a significant role in regulation of matrix metalloproteinase (MMP) expression [15]. MMPs, also known as matrixins, are produced through selective degradation of the extracellular matrix (ECM), which is composed of structural proteins such as collagens, proteoglycans, and glycoproteins. MMPs are involved in organogenesis, nerve growth, ovulation, uterine dilatation, bone formation, wound healing, and angiogenesis. MMPs are also suggested to play an important role in embryogenesis and the implantation process. MMP-2 and MMP-9 are produced by human trophoblast cells [16], and cultured embryos have also been reported to secrete MMP-2 [17]. Recent studies suggest that MMPs not only play a major role in ovarian ECM reorganization 3 [18] but are also associated with RPL [8,19].

In our previous studies, a single nucleotide variation in *HOTAIR* was found to be associated with pathology in patients with recurrent implantation failure [20], primary ovarian insufficiency [21], and colorectal cancer [22]. *HOTAIR* is well known for occurring alongside cancer, and its polymorphisms are also known as various diseases; selected polymorphisms (rs920778, rs1899663, rs4709314, and rs12826786) are especially associated and reported with certain diseases [23,24,25,26]. Polymorphisms are associated with gene expression; *HOTAIR* SNPs also regulate gene expression [25]. Finally, the lncRNA *HOTAIR* has been reported to regulate CCND1 and CCND2 expressions [24]; CCND1 has been identified as an important factor in oocyte development and meiotic maturation and may serve as a potential target for improving in vitro fertilization outcomes [26]. In this study, we investigated four *HOTAIR* polymorphisms associated with recurrent pregnancy loss and how these SNPs change risk of RPL. *HOTAIR* polymorphisms are already reported in various diseases, but this is the first report in which they were associated with RPL. To investigate the relationship between RPL and *HOTAIR* SNVs, we recruited RPL patients and healthy controls.

## 2. Materials and Methods

### 2.1. Study Population

A total of 403 women with RPL (mean age: 30.93 ± 4.13 years) and 383 healthy women (mean age: 32.83 ± 4.17 years) without reported fertility complications were recruited between March 1999 and February 2012 by the Department of Obstetrics and Gynecology of CHA Bundang Medical Center, CHA University (Seongnam, Republic of Korea). Women with RPL were defined as those with a history of at least two consecutive spontaneous miscarriages before 20 weeks of gestation. All control participants had regular menstrual cycles, normal nuclear type 46XX, and a history of at least one natural birth with healthy conditions. Patients with any history of pelvic surgery, cancer, radiation exposure, autoimmune disorder, or genetic syndrome were excluded from this study. Participants who had a history of smoking or alcohol use were also excluded from this study. Baseline blood tests were performed to examine typical miscarriage causes, such as thyroid diseases and hyperprolactinemia.

The institutional review board of the CHA Bundang Medical Center reviewed and approved this study, and written informed consent was obtained from all participants (IRB number: 2010-01-123). All participants were informed of the study protocols and signed an informed consent form before participating in this study.

### 2.2. Genotyping

DNA was extracted from leukocytes via the G-DEX™ II Genomic DNA Extraction kit (Intron Biotechnology, Seongnam, Korea), according to the manufacturer’s instructions, and a total of four *HOTAIR* polymorphisms were analyzed. All PCR experiments were performed using AccuPower HotStart PCR PreMix (Bioneer Corporation, Daejeon, Korea).

*HOTAIR* polymorphisms rs1899663, rs4759314, and rs920778 were identified by polymerase chain reaction restriction fragment length polymorphism (PCR-RFLP) analysis, whereas rs7958904 was identified using real-time PCR. Primers and TaqMan probes were designed with Primer Express Software (version 2.0; Thermo Fisher Scientific, Inc., Waltham, MA, USA) and synthesized by Applied Biosystems (Foster City, CA, USA). All PCR experiments were performed using AccuPower HotStart PCR PreMix (Bioneer Corporation, Daejeon, Korea).

Polymorphism genotypes rs920778, rs1899663, and rs4759314 were determined by PCR-RFLP, followed by digestions with restriction enzymes MspI, HphI, and AluI (New England BioLabs, Inc., Ipswich, MA, USA), respectively. All restriction digests were performed for 16 h at the optimal temperature for each enzyme’s activity (37 °C for HphI, 55 °C for AluI, and 37 °C for MspI). Digestion products were separated by electrophoresis on a 4% agarose gel and visualized with ethidium bromide using a Gel Doc XR+ version system (BioRad, Hercules, CA, USA).

The following primers were used to amplify the rs1899663 polymorphism: forward, 5′-TTT TCC AGT TGA GGA GGG TGG A-3′; and reverse, 5′-CTA ATG GCA AGG GAA GGG AAG G-3′. Digestion of the 114 bp PCR product with *Hph*I resulted in one fragment for the homozygous GG genotypes (114 bp), two fragments for the homozygous TT genotype (79 bp and 35 bp), and three fragments for the heterozygous GT genotype (114 bp, 79 bp, and 35 bp).

The following primers were used to amplify the rs4759314 polymorphism: forward, 5′-ACC CAA AAC CAT TTC CTG AGA G-3′; and reverse, 5′-TTC AGG TTT TAT TAA CTT GCA TCA GC-3′. Digestion of the 124 bp PCR product with *Alu*I resulted in a single fragment for the homozygous GG genotype (124 bp), two fragments for the homozygous AA genotype (98 bp and 25 bp), and three fragments for the heterozygous GA genotype (124 bp, 99 bp, and 25 bp).

The following primers were used to amplify the rs920778 polymorphism: forward, 5′-GCC TCT GGA TCT GAG AAA GAA A-3′; and reverse, 5′-TTA CAG CTT AAA TGT CTG AAT GTT CC-3′. Digestion of the 140 bp PCR product with *Msp*I resulted in one fragment for the homozygous TT genotype (140 bp), two fragments for the homozygous CC genotype (113 bp and 27 bp), and three fragments for the heterozygous TC genotype (140 bp, 113 bp, and 27 bp).

Real-time PCR was performed with an RG-6000 system (Corbett Research, Australia), using reporter dyes 5-carboxyfluorescein (FAM) and 2′, 7′-dimethoxy-4′, and 5′-dichloro-6-carboxyfluorescein (JOE), as well as dark quencher black-hole quencher 1 (BHQ1). The probe used to detect the rs7958904 C allele was 5′-[JOE]-CG GCT CGG GTC AG-[BHQ1]-3′, whereas the probe used to detect the rs7958904 G allele was 5′-[FAM]-CG GCT CCG GTC AG-[BHQ1]-3′.

To validate the PCR-RFLP findings for each polymorphism, 30% of the PCR assays were randomly selected and duplicated, followed by DNA sequencing using the ABI 3730xl DNA Analyzer (Applied Biosystems). The concordance of the quality control samples was 100%.

### 2.3. Clinical Assessment

Blood samples from 403 RPL patients were collected during pregnancy. Blood samples were collected by venipuncture on day 2 or 3 of the menstrual cycle for measurement of FSH, luteinizing hormone (LH), and estradiol (E2). Serum was prepared as previously described [27], and hormone levels were determined using either radioimmunoassay (E2 [cat. no., A21854], Beckman Coulter, Inc., Brea, CA, USA) or enzyme immunoassay on the IMMULITE^®^ 1000 System (FSH and LH; Siemens AG, Munich, Germany). Plasma homocysteine, folate, total cholesterol, urate concentrations, and blood coagulation factors were measured in RPL patients after they had fasted for 12 h. Homocysteine levels were measured using a fluorescence polarization immunoassay and an Abbott IMx analyzer (Abbott Laboratories, Abbott Park, IL, USA). Folate levels (15.31 ± 14.8 ng/mL) were determined using a competitive immunoassay with ACS:180 (Bayer Diagnostics, Tarrytown, NY, USA). Total cholesterol (187.40 ± 48.96 mg/dL) and urate levels (3.79 ± 0.83 mg/dL) were determined with enzymatic colorimetric assay kits (Roche Diagnostics, Mannheim, Germany). Platelet (PLT) counts (255.21 ± 59.34 103 cells/μL) were measured using a Sysmex XE2100 automated hematology analyzer (Sysmex, Kobe, Japan), and PT (11.31 ± 1.77 s) and aPTT (32.01 ± 4.28 s) were measured using an automated photooptical coagulometer (ACL TOP; Mitsubishi Chemical Medience, Tokyo, Japan) to assess blood coagulation parameters.

### 2.4. Statistical Analysis

Differences in the genotypic frequencies of the four *HOTAIR* polymorphisms between the control group and patients with RPL were assessed using Fisher’s exact test and logistic regression. Odds ratios (ORs), adjusted ORs (AORs), and 95% confidence intervals (CIs) were also calculated. Data analysis was conducted using MedCalc version 12.1.4 (MedCalc, Ostend, Belgium) and GraphPad Prism 4.0 (GraphPad, San Diego, CA, USA). Allele frequencies were calculated to investigate deviations from Hardy–Weinberg equilibrium (HWE). Gene–gene interactions among the four lncRNA loci were analyzed using the multifactor dimensionality reduction (MDR) method in MDR software version 2.0 (www.epistasis.org) (accessed on 6 October 2022). (released in 2008) [28]. Using the MDR method, we evaluated the frequencies of all combinations of the four polymorphisms, followed by Fisher’s exact test to adjust for multiple comparisons. Allelic frequencies were assessed for HWE, using *p* < 0.05 as the significance threshold. Haploview 4.2 was used to calculate predicted heterozygosity and Hardy–Weinberg equilibrium and to estimate linkage disequilibrium haplotype blocks in the *HOTAIR* gene region. Differences in hormone concentrations (estradiol (E2), follicle stimulating hormone (FSH), luteinizing hormone (LH), prolactin, and thyroid stimulating hormone (TSH)) associated with different *HOTAIR* genotypes and alleles were evaluated using a one-way analysis of variance with a post hoc Scheffé test for all pairwise comparisons or with an independent two-sample Student’s *t*-test, as appropriate. Data are presented as the mean ± standard deviation.

## 3. Results

### 3.1. Patient Clinical Characteristics

Patients’ clinical characteristics, including body mass index; number of previous pregnancy losses; mean gestational age; percent CD56+ natural killer cells; and levels of PT, homocysteine, folate, total cholesterol, uric acid, and plasminogen activator inhibitor (PAI)-1, are listed in Table 1. No significant difference in mean age was identified between patients with RPL and the control group. Importantly, compared with the control group, patients with RPL had significantly increased levels of E2 (*p* < 0.0001), PT (*p* = 0.0002), and aPTT (*p* < 0.0001), as well as decreased levels of total cholesterol (*p* < 0.0001).

### 3.2. Genetic Analysis

Table 2 and Appendix A show the genotype and allele frequencies of the four *HOTAIR* lncRNAs in both RPL patients and the control group. The *HOTAIR* rs4759314 A>G and rs920778 T>C polymorphisms were significantly more prevalent in patients with RPL than in the control group (Table 2 and Appendix A), suggesting association of these polymorphisms with increased RPL risk. However, *HOTAIR* rs920778 did not remain significantly associated with RPL risk after adjustment using the false discovery rate correction (*p* = 0.076). The *HOTAIR* genotype frequencies observed in the RPL and control groups were consistent with the expected HWE values.

Haplotype analyses of the four *HOTAIR* polymorphisms are shown in Table 3 and Appendix A. Among the analyzed haplotypes, eight were more frequent in patients with RPL than in the control group, including A-T-G-C, A-T-T-G, A-C-G-G, A-C-T-G, G-T-G-G, G-T-T-G, G-T-T-C, and G-C-T-G. Among the models that examined three *HOTAIR* polymorphic loci, the rs4759314/rs920778/rs1899663 haplotypes A-T-T, A-C-G, G-T-G, G-T-T, and G-C-T; the rs4759314/rs1899663/rs7958904 haplotypes A-G-C, A-T-G, G-G-G, G-T-G, and G-T-C; and the rs920778/rs1899663/rs7958904 haplotypes T-G-C, T-T-G, C-G-G, and C-T-G (Table 3 and Appendix A) were significantly associated with increased RPL risk. By contrast, *HOTAIR* polymorphic haplotypes A-T-C (OR, 0.161; 95% CI, 0.083–0.309; *p* < 0.0001) and G-T-C (OR, 0.051; 95% CI, 0.002–0.881; *p* = 0.002) for rs4759314/rs920778/rs7958904 were associated with decreased RPL risk. Among the models that examined two *HOTAIR* polymorphic loci, rs4759314/rs920778 haplotypes G-G and G-T, rs4759314/rs7958904 haplotype G-G, rs920778/rs1899663 haplotype C-G, and rs1899663/rs7958904 haplotypes G-C and T-G were associated with increased RPL risk.

Table 4 and Appendix A show the results of the combined genotypic analysis. The following *HOTAIR* genotypes were associated with increased RPL risk: rs4759314/rs920778 genotypes AA/CC, AG/TT, and AG/TC; rs4759314/rs1899663 genotypes AG/GG, AG/GT, and AG/TT; rs4759314/rs7958904 genotype AG/GG; rs920778/rs1899663 genotype CC/GG; rs920778/rs7958904 genotypes TC/GG and CC/GG; and rs1899663/rs7958904 genotypes GG/GC and GT/GG. These results are consistent with associations between increased RPL risk and individual *HOTAIR* genotypes.

The results of linkage disequilibrium (LD) are shown in the heatmap (Figure 1). In the control, the r2 values between rs920778 and rs1899663, rs920778 and rs4759314, and rs1899663 and rs4759314 were 0.65, 0.93, and 0.77, respectively. In case studies, the r2 values between rs920778 and rs1899663, rs920778 and rs4759314, and rs1899663 and rs4759314 were 0.49, 0.64, and 0.42, respectively.

Differences in plasma levels of clinical factors associated with RPL risk among the various *HOTAIR* genotypes associated with polymorphic loci were also evaluated, including uric acid, homocysteine, FSH, and PRL, as shown in Appendix A. Patients with the *HOTAIR* rs7958904 CC genotype had significantly lower CD19 levels than patients with the *HOTAIR* rs7958904 GG and GC genotypes (*p* = 0.032; Appendix A, Appendix A). Abnormal *HOTAIR* expression has also been reported in development of various diseases [29,30,31].

## 4. Discussion

We evaluated the association between *HOTAIR* polymorphisms and RPL risk. To the best of our knowledge, this is the first study to investigate the relationship between polymorphisms in the lncRNA *HOTAIR*—including rs4759314 A>G, rs920778 T>C, rs1899663 G>T, and rs7958904 G>C—and susceptibility to RPL in a Korean population. Our results indicate that the *HOTAIR* rs4759314 [15] and rs920778 [16] polymorphisms may be associated with an increased risk of RPL.

*HOTAIR* is a well-known lncRNA. The 5′ end of the fragment interacts with the polycomb repressive complex 2 (PRC2), which is associated with histone methyltransferase activity, whereas the 3′ end interacts with the lysine specific demethylase 1 (LSD1) gene, which is involved in development of hematopoietic stem cells. Aberrant *HOTAIR* expression has previously been reported to play a role in the development of estrogen receptor-positive breast cancer [32]. *HOTAIR* activated the expression of proteins involved in nuclear factor kappa B (NF-κB) [33] and associated with increased tumor cell growth rates in liver cancer [34]. Single-nucleotide variants, such as rs920778 and rs12826786, regulate *HOTAIR* expression.

We evaluated the association between four *HOTAIR* polymorphisms (rs4759314, rs920778, rs1899663, and rs7958904) and RPL development in a population of Korean women. Our results demonstrate that *HOTAIR* rs4759314/rs920778 genotypes AA/CC and AG/TT, rs1899663/rs7958904 genotypes GG/GC and GT/GG, rs4759314/rs920778 haplotype G-T, and rs1899663/rs7958904 haplotypes G-C and T-G are associated with increased RPL frequency. Patients with the *HOTAIR* rs4759314 G allele exhibited a significant increase in RPL, whereas patients with the *HOTAIR* rs4759314 A allele showed a decreased prevalence of RPL.

*HOTAIR* has been suggested to play roles in oncogenesis [35], heart disease [36], and nervous system functions [37]. *HOTAIR* has been proposed as a possible biomarker for cardiovascular diseases [38]. A feature of the long non-coding RNA, the *HOTAIR* gene is bound to other microRNA or other genes and regulates expression. The *HOTAIR* gene is indirectly regulated by CCND1 and CCND2 gene expression and binding to microRNA [39]. The CCND1 gene is an especially important factor of meiotic maturation and oocyte development [40]. Therefore, we speculated that *HOTAIR* is related to RPL occurrence.

In this study, we performed a detailed analysis of *HOTAIR* variants and found that patients harboring the *HOTAIR* rs4759314 GG genotype had higher platelet counts than patients with the AA genotype. We propose that this *HOTAIR* polymorphism can serve as a potential biomarker for RPL susceptibility.

In our results, the patient group and the control group showed differences in TSH levels. However, in Appendix A, there is no significant difference observed in genotypes and TSH levels. Many papers state that TSH levels are associated with early pregnancy loss [41]. Therefore, we need further study of association between *HOTAIR* expression and TSH levels.

Our research had some limitations. First, the mechanisms through which detected polymorphisms in *HOTAIR* affect onset of RPL remain unknown and must be elucidated by further studies. Second, the CIs around the ORs were overly broad, reflecting the small sample size, which is an important limitation in association studies. Third, information regarding other risk factors for developing RPL is lacking, and functional analysis of these factors is needed. Fourth, we could not examine the expression level of each genotype. We need further functional studies of direct effect of genotype and *HOTAIR* expression; however, there are several reports that genotypes of *HOTAIR* are associated with *HOTAIR* expression [26,42].

In summary, we found that the *HOTAIR* rs4759314 AG and AG + GG genotypes and the *HOTAIR* rs920778 CC and TT + TC genotypes were more frequently identified among patients with RPL than were the wild-type genotypes rs4759314 AA and rs920778 TT. This finding suggests that wild-type *HOTAIR* may confer a potential protective effect against RPL. To the best of our knowledge, this is the first study to evaluate the association between *HOTAIR* polymorphisms (rs4759314 A>G, rs920778 T>C, rs1899663 G>T, and rs7958904 G>C) and RPL in a Korean population.

## Figures and Tables

**Figure 1 genes-13-02138-f001:**
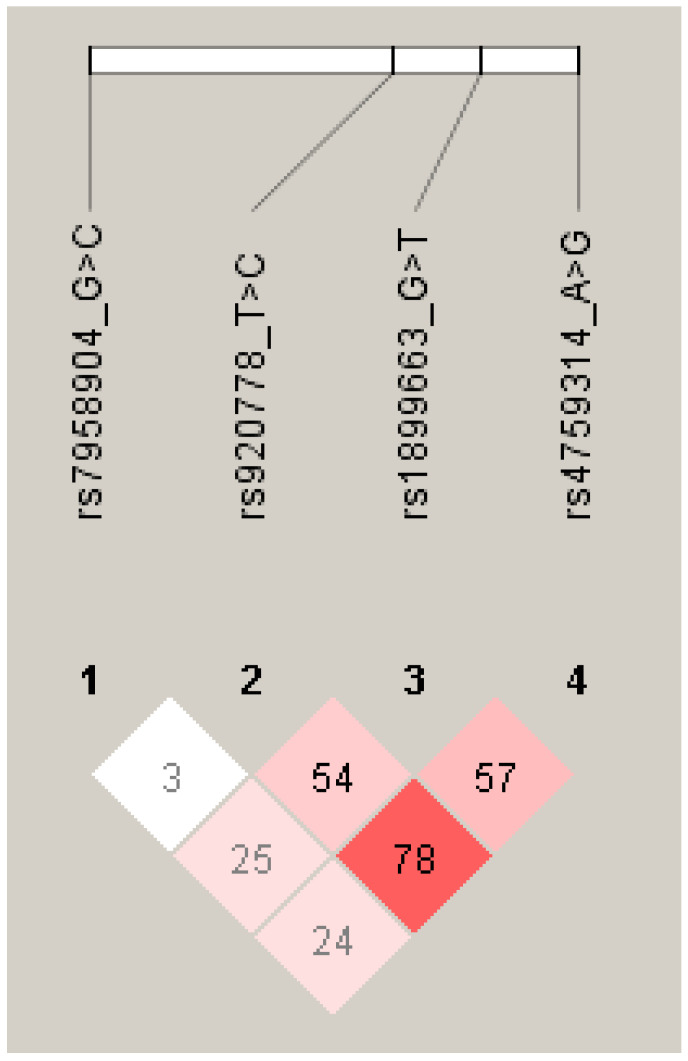
Linkage disequilibrium between *HOTAIR* loci.

**Table 1 genes-13-02138-t001:** Baseline characteristics between patients with recurrent pregnancy loss (RPL) and controls.

Characteristics	Controls (*n* = 383)	RPL (*n* = 403)	*p* ^a^
Age (years, mean ± SD)	32.83 ± 4.17	32.93 ± 4.13	0.822
BMI (kg/m^2^)	21.63 ± 3.26	21.48 ± 3.89	0.738 ^b^
Live birth (n, mean ± SD)	1.65 ± 0.58	N/A	
Pregnancy loss (n, mean ± SD)	N/A	2.99 ± 1.48	
Mean gestational age (weeks)	39.29 ± 1.59	N/A	
PLT (10^3^ /µL)	239.09 ± 64.05	244.84 ± 62.79	0.258
PT (s)	10.78 ± 1.71	11.34 ± 1.66	0.0002
aPTT (s)	29.34 ± 3.60	31.95 ± 4.36	<0.0001 ^b^
Folate (ng/mL)	14.07 ± 9.72	15.20 ± 14.75	0.934 ^b^
Homocysteine (µmol/L)	7.40 ± 5.13	6.91 ± 2.07	0.858 ^b^
Total cholesterol (mg/dL)	212.15 ± 57.77	187.18 ± 46.84	<0.0001 ^b^
Uric acid (mg/dL)	3.82 ± 1.02	3.83 ± 0.87	0.742 ^b^
BUN (mg/dL)	8.95 ± 3.04	10.37 ± 6.24	<0.0001 ^b^
Creatinine (mg/dL)	0.64 ± 0.16	0.73 ± 0.12	<0.0001 ^b^
FSH (mIU/mL)	8.12 ± 3.11	7.72 ± 11.43	<0.0001 ^b^
LH (mIU/mL)	3.59 ± 2.55	6.31 ± 11.90	<0.0001 ^b^
E2 (pg/mL)	26.00 ± 14.75	52.37 ± 119.94	<0.0001 ^b^
TSH (µIU/mL)	1.55 ± 1.04	2.14 ± 1.44	<0.0001 ^b^
LDL cholesterol (mg/dL)	145.39 ± 41.65	109.62 ± 35.41	0.0001
HDL cholesterol (mg/dL)	84.31 ± 20.32	59.94 ± 15.82	<0.0001
Triglyceride (mg/dL)	181.26 ± 110.32	184.22 ± 145.73	0.299^b^
Glucose (mg/dL)	100.05 ± 24.04	96.58 ± 13.54	0.893 ^b^
FBS (mg/dL)	6.55 ± 1.96	95.07 ± 16.83	<0.0001 ^b^
WBC (10^3^/µL)	7.99 ± 2.52	7.32 ± 2.83	0.006
RBC (10^6^/µL)	4.09 ± 0.39	4.22 ± 0.42	0.0003
Hemoglobin (g/dL)	12.21 ± 1.24	12.61 ± 1.33	0.0006
Hematocrit (%)	36.80 ± 3.35	37.57 ± 3.75	0.005 ^b^
PAI-1 (ng/mL)	N/A	11.30 ± 7.81	
CD3 (%)	N/A	67.33 ± 8.84	
CD4 (%)	N/A	36.42 ± 7.34	
CD8 (%)	N/A	27.99 ± 7.61	
CD19 (%)	N/A	12.54 ± 4.86	
CD56 NK cells (%)	N/A	17.66 ± 7.88	

^a^ Two-sided *t*-test. ^b^ Mann–Whitney test. BMI, body mass index; PLT, platelet; PT, prothrombin time; aPTT, activated partial thromboplastin time; BMI, body mass index; BUN, blood urea nitrogen; FSH, follicle-stimulating hormone; LH, luteinizing hormone; E2, estradiol; TSH, thyroid-stimulating hormone; HDL cholesterol, high-density lipoprotein cholesterol; FBS, fasting blood sugar; PAI-1, Plasminogen activator inhibitor-1; CD3, cluster of differentiation 3; SD, standard deviation.

**Table 2 genes-13-02138-t002:** Comparison of genotype frequencies and AOR values of polymorphism between the RPL and control subjects.

Genotypes	Controls (*n* = 383)	RPL (*n* = 403)	AOR (95% CI)	*p*	FDR
***HOTAIR* rs4759314**					
AA	351 (91.6)	322 (80.0)	1.000 (reference)		
AG	30 (7.8)	79 (19.6)	3.054 (1.847-5.050)	<0.0001	<0.0001
GG	2 (0.5)	2 (0.5)	1.095 (0.098-12.175)	0.941	0.941
Dominant (AA vs. AG+GG)			2.933 (1.792-4.801)	<0.0001	<0.0001
Recessive (AA+AG vs. GG)			0.947 (0.085-10.522)	0.965	0.965
HWE-P	0.134	0.22			
A allele			1.000 (reference)		
G allele			2.479 (1.642–3.744)	<0.0001	
***HOTAIR* rs920778**					
TT	236 (61.6)	226 (56.1)	1.000 (reference)		
TC	134 (35.0)	149 (37.0)	1.166 (0.867–1.568)	0.310	0.620
CC	13 (3.4)	28 (6.9)	2.263 (1.140–4.488)	0.019	0.076
Dominant (TT vs. TC+CC)			1.264 (0.950–1.680)	0.107	0.214
Recessive (TT+TC vs. CC)			2.129 (1.085–4.175)	0.028	0.112
HWE-P	0.251	0.612			
T allele			1.000 (reference)		
C allele			1.297 (1.025–1.641)	0.031	
***HOTAIR* rs1899663**					
GG	224 (58.5)	235 (58.3)	1.000 (reference)		
GT	139 (36.3)	146 (36.2)	0.921 (0.645–1.316)	0.651	0.756
TT	20 (5.2)	22 (5.5)	0.756 (0.331–1.726)	0.506	0.675
Dominant (GG vs. GT+TT)			0.902 (0.640–1.273)	0.558	0.558
Recessive (GG+GT vs. TT)			0.800 (0.357–1.794)	0.589	0.589
HWE-P	0.794	0.913			
G allele			1.000 (reference)		
T allele			0.987 (0.788–1.247)	0.915	
***HOTAIR* rs7958904**					
GG	210 (54.8)	232 (57.6)	1.000 (reference)		
GC	144 (37.6)	144 (35.7)	0.945 (0.662–1.349)	0.756	0.756
CC	29 (7.6)	27 (6.7)	0.678 (0.326–1.413)	0.300	0.600
Dominant (GG vs. GC+CC)			0.896 (0.638–1.260)	0.529	0.558
Recessive (GG+GC vs. CC)			0.682 (0.333–1.397)	0.296	0.296
HWE-P	0.534	0.471			
G allele			1.000 (reference)		
C allele			0.914 (0.728–1.146)	0.435	

^a^ Fisher’s exact test. ^b^ False discovery rate-adjusted *p* value for multiple hypotheses testing using the Benjamini-Hochberg method. Acceptance of statistical significance at *p* < 0.05 and 95% CI not including 1. Note: For AOR was adjusted by age of participants. RPL = recurrent pregnancy loss; AOR = adjusted odds ratio; CI = confidence interval.

**Table 3 genes-13-02138-t003:** Haplotype analysis of *HOTAIR* polymorphisms in RPL and controls subjects.

Haplotype	Controls (2n = 766)	Case (2n = 806)	OR (95% CI)	*p* *
rs4759314A>G/rs920778 T>C/rs1899663 G>T/rs7958904 G>C				
A-T-G-G	0.689	0.577	1.000 (reference)	
A-T-G-**C**	0.007	0.039	7.040 (2.715–18.260)	<0.0001
A-T-**T**-G	0.005	0.025	5.677 (1.926–16.730)	0.001
A-**C**-G-G	0.025	0.071	3.406 (1.997–5.811)	<0.0001
A-**C**-G-**C**	0.002	0.010	4.542 (0.959–21.500)	0.053
A-**C**-**T**-G	0.006	0.015	3.406 (1.091–10.640)	0.040
**G**-T-G-G	0.009	0.036	4.704 (2.041–10.840)	<0.0001
**G**-T-**T**-G	0.000	0.005	10.220 (0.548–190.400)	0.049
**G**-T-**T**-**C**	0.001	0.012	11.350 (1.447–89.080)	0.004
**G**-**C**-**T**-G	0.000	0.016	30.650 (1.816–517.400)	<0.0001
rs4759314A>G/rs920778 T>C/rs1899663 G>T				
A-T-G	0.699	0.614	1.000 (reference)	
A-**C**-G	0.028	0.079	3.255 (1.957–5.415)	<0.0001
**G**-T-G	0.015	0.044	3.551 (1.788–7.054)	<0.0001
**G**-T-**T**	0.005	0.017	3.798 (1.241–11.620)	0.016
**G**-**C**-**T**	0.000	0.013	24.950 (1.466–424.900)	<0.0001
rs4759314A>G/rs920778 T>C/rs7958904 G>C				
A-T-G	0.683	0.754	1.000 (reference)	
A-T-**C**	0.077	0.013	0.161 (0.083–0.309)	<0.0001
**G**-T-**C**	0.011	0.000	0.051 (0.002–0.881)	0.002
rs4759314A>G/rs1899663 G>T/rs7958904 G>C				
A-G-G	0.714	0.648	1.000 (reference)	
A-G-**C**	0.009	0.048	5.827 (2.583–13.150)	<0.0001
A-**T**-G	0.011	0.041	4.314 (1.974–9.429)	<0.0001
**G**-G-G	0.011	0.043	4.067 (1.936–8.546)	<0.0001
**G**-**T**-G	0.000	0.021	36.600 (2.194–610.700)	<0.0001
**G**-**T**-**C**	0.002	0.014	11.500 (1.479–89.470)	0.003
rs920778 T>C/rs1899663 G>T/rs7958904 G>C				
T-G-G	0.697	0.613	1.000 (reference)	
T-G-**C**	0.016	0.046	3.077 (1.616–5.857)	<0.0001
T-**T**-G	0.006	0.031	6.756 (2.334–19.560)	<0.0001
**C**-G-G	0.028	0.079	3.145 (1.908–5.183)	<0.0001
**C**-**T**-G	0.005	0.032	6.756 (2.334–19.560)	<0.0001
rs4759314A>G/rs920778 T>C				
A-T	0.767	0.685	1.000 (reference)	
**G**-T	0.024	0.061	2.742 (1.594–4.718)	<0.0001
**G**-**C**	0.020	0.042	2.410 (1.298–4.475)	0.004
rs4759314A>G/rs1899663 G>T				
A-G	0.725	0.694	1.000 (reference)	
**G**-G	0.042	0.071	1.769 (1.129–2.770)	0.012
**G**-**T**	0.003	0.032	12.910 (3.048–54.660)	<0.0001
rs4759314A>G/rs7958904 G>C				
A-G	0.723	0.686	1.000 (reference)	
**G**-G	0.013	0.069	5.510 (2.780–10.920)	<0.0001
rs920778 T>C/rs1899663 G>T				
T-G	0.711	0.657	1.000 (reference)	
**C**-G	0.055	0.108	2.134 (1.449–3.144)	<0.0001
rs920778 T>C/rs7958904 G>C				
T-G	0.702	0.641	1.000 (reference)	
**C**-G	0.034	0.113	3.642 (2.317–5.726)	<0.0001
rs1899663 G>T/rs7958904 G>C				
G-G	0.725	0.691	1.000 (reference)	
G-**C**	0.041	0.073	1.900 (1.211–2.981)	0.005
**T**-G	0.011	0.063	6.364 (2.992–13.540)	<0.0001

Note: OR, odds ratio; CI, confidence interval.

**Table 4 genes-13-02138-t004:** Combined genotype analysis for the *HOTAIR* polymorphisms rs4759314A>G, rs920778T>C, rs1899663G>T, and rs7958904G>C in RPL patients and controls.

Genotype Combination	Controls (*n* = 383)	RPL (*n* = 403)	AOR (95% CI)	*p **
rs4759314/rs920778				
AA/TT	223 (58.2)	190 (47.1)	1.000 (reference)	
AA/CC	10 (2.6)	20 (5.0)	2.415 (1.100–5.302)	0.028
AG/TT	11 (2.9)	35 (8.7)	3.778 (1.865–7.652)	0.0002
AG/TC	16 (4.2)	36 (8.9)	2.690 (1.445–5.006)	0.002
rs4759314/rs1899663				
AA/GG	199 (52.0)	191 (47.4)	1.000 (reference)	
AG/GG	23 (6.0)	43 (10.7)	1.964 (1.139–3.389)	0.015
AG/GT	6 (1.6)	28 (6.9)	4.798 (1.941–11.861)	0.0007
AG/TT	1 (0.3)	8 (2.0)	8.529 (1.056–68.916)	0.044
rs4759314/rs7958904				
AA/GG	203 (53.0)	188 (46.7)	1.000 (reference)	
AG/GG	6 (1.6)	44 (10.9)	7.874 (3.277–18.920)	<0.0001
rs920778 / rs1899663				
TT/GG	197 (51.4)	186 (46.2)	1.000 (reference)	
CC/GG	7 (1.8)	22 (5.5)	3.387 (1.411–8.134)	0.006
rs920778/rs7958904				
TT/GG	197 (51.4)	183 (45.4)	1.000 (reference)	
TC/GG	5 (1.3)	22 (5.5)	4.792 (1.776–12.933)	0.002
CC/GG	8 (2.1)	27 (6.7)	3.614 (1.598–8.173)	0.002
rs1899663/rs7958904				
GG/GG	202 (52.7)	193 (47.9)	1.000 (reference)	
GG/GC	21 (5.5)	35 (8.7)	1.826 (1.023–3.261)	0.042
GG/CC	1 (0.3)	7 (1.7)	7.735 (0.939–63.701)	0.057
GT/GG	8 (2.1)	34 (8.4)	4.293 (1.932–9.538)	0.0003

* The odds ratio was adjusted by age. RPL, recurrent pregnancy loss; AOR, adjusted odds ratio; 95% CI, 95% confidence interval.

## Data Availability

Data sharing not applicable.

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
