# Peer review of "Association of Polymorphisms in the Long Non-Coding RNA HOTAIR with Recurrent Pregnancy Loss in a Korean Population"

_genes, 2022, doi:10.3390/genes13112138_

Round 1

Reviewer 1 Report

In this study, the authors investigated HOTAIR polymorphisms (rs4759314 A>G, rs920778 T>C, rs1899663 G>T, and rs7958904 G>C) and the risk of RPL in Korean women. Although this topic is interesting, and the study is well-designed, I have several concerns to be addressed as follows:

1-     Introduction:

-        Authors should mention the location of the gene encoding HOTAIR.

-        There are several genetic variants of HOTAIR. Please describe them and explain why only these four SNPs were chosen for patients with RPL.

-        The hypothesis of the study as well as the objectives should be defined clearly in the last paragraph of this section.

2-     Methods:

-        Several clinical, hormonal, and immunological assays were performed, such as FSH, LH, E2, TSH, PAI-1, CD 3, 4, 8, 19, etc. Please define these assays in the methods.

-        The Inc HOTAIR relative expression level should be assayed and compared among different genotypes to determine the impact of HOTAIR SNPs on its expression level.

3-     Results:

-        Many factors could affect the occurrence of RPL, such as thrombophilia, endocrine disorders, immune dysfunction, smoking, etc. which are estimated in cases and controls. However, they should be used to adjust the OR of each SNP.

-        A multiplicative genetic model should be assessed (allelic model).

4-     Throughout the manuscript, there are several structural and grammatical errors. The whole manuscript should be carefully revised by a native English speaker.

Author Response

1-     Introduction:

-        Authors should mention the location of the gene encoding HOTAIR.

Thank you for your comment. We are added that our opinion in line 51 as follow

HOTAIR is located in HOX C cluster genes which is located in chromosome 12 and regulate HOXD genes

-        There are several genetic variants of HOTAIR. Please describe them and explain why only these four SNPs were chosen for patients with RPL.

Thank you for your comment. We are added the explain in line 63

HOTAIR is well known for occur cancer and their polymorphisms are also known as various disease especially selected polymorphisms (rs920778, rs1899663, rs4709314 and rs12826786) are associated reported with disease.

-        The hypothesis of the study as well as the objectives should be defined clearly in the last paragraph of this section.

Thank you for your comment. We are added the explain in line 69

In this study we investigated four HOTAIR polymorphisms are associated with recurrent pregnancy loss. And how these SNPs are change risk of RPL. HOTAIR polymorphisms are already reported with various disease but this is the first report that associated with RPL. To investigate the relationship between RPL and HOTAIR SNVs. We recruit RPL patients and healthy controls.

2-     Methods:

-        Several clinical, hormonal, and immunological assays were performed, such as FSH, LH, E2, TSH, PAI-1, CD 3, 4, 8, 19, etc. Please define these assays in the methods.

Thank you for your comment. We are added the explain in line 130

Blood samples were collected by venipuncture on day 2 or 3 of the menstrual cycle for measurement of FSH, luteinizing hormone (LH), and estradiol (E2). Serum was prepared as previously described [26], and hormone levels were determined using either radioimmunoassay (E2 [cat. no., A21854], Beckman Coulter, Inc., Brea, CA, USA) or enzyme immunoassay on the IMMULITE® 1000 System (FSH and LH; Siemens AG, Munich, Germany).

-        The Inc HOTAIR relative expression level should be assayed and compared among different genotypes to determine the impact of HOTAIR SNPs on its expression level.

Thank you for your comment. We are added the explain in line 250

Fourth, we couldn’t examine the expression level of each genotype. We need to further study for functional studies to directly effect genotype and HOTAIR expression however, there are several report that genotype of HOTAIR are associated with HOTAIR expression[25,40].

3-     Results:

-        Many factors could affect the occurrence of RPL, such as thrombophilia, endocrine disorders, immune dysfunction, smoking, etc. which are estimated in cases and controls. However, they should be used to adjust the OR of each SNP.

Thank you for your comment. We are added the explain in line 84

Participants who had a history of smoking or alcohol use were excluded from this study. Baseline blood tests were performed to examine typical miscarriage causes, such as thyroid diseases and hyperprolactinemia.

-        A multiplicative genetic model should be assessed (allelic model).

Thank you for your comment. We added allele type in table 1.

4-     Throughout the manuscript, there are several structural and grammatical errors. The whole manuscript should be carefully revised by a native English speaker.

Thank you for your comment. we will revise grammatical errors

Reviewer 2 Report

The manuscript is interesting but needs small corrections

Please add  A total of 10-12% in the introduction section Line 38

Rewrite the sentence RPL development frequencies Line 206 

Please specify the most prevalent risk haplotype resulting in RPF in Korean population along with possible mechanism in the etiology of RPL.

Author Response

The manuscript is interesting but needs small corrections

Please add  “A total of 10-12%” in the introduction section Line 38

Thank you for your comment. We are added “A total” in line 38

Rewrite the sentence RPL development frequencies Line 206 

Thank you for your comment. We are rewrite in line 206

By contrast, the HOTAIR polymorphic haplotypes A-T-C (OR, 0.161; 95% CI, 0.083–0.309; P < 0.0001) and G-T-C (OR, 0.051; 95% CI, 0.002–0.881; P = 0.002) for rs4759314/rs920778/rs7958904 were associated with decreased RPL risk.

Please specify the most prevalent risk haplotype resulting in RPF in Korean population along with possible mechanism in the etiology of RPL.

Thank you for your comment. We are rewrite in line 238

Feature of the long non-coding RNA, HOTAIR gene is bind with other microRNA or other genes and regulate the expression, HOTAIR gene is indirectly regulated with CCND1 and CCND2 gene expression to bind with microRNA. Especially CCND1 gene is one of the important factor of meiotic maturation and oocyte development. So we speculate HOTAIR related with RPL occurrence.

Round 2

Reviewer 1 Report

The authors have adequately addressed most of my concerns. However, the whole manuscript should be carefully revised by a native English speaker as there are several structural and grammatical errors that make the manuscript hard to read.